immunology, cellular biology, systems biology

macrophage, cannibalism, efferocytosis, inflammation, mathematical biology, coagulation – fragmentation

**Author for correspondence:**
David R. Greaves
e-mail: david.greaves@path.ox.ac.uk

# Efferocytosis perpetuates substance accumulation inside macrophage populations

Hugh Z. Ford[1,2,4], Lynda Zeboudj[1], Gareth S. D. Purvis[1], Annemieke ten Bokum[1], Alexander E. Zarebski[3], Joshua A. Bull[2], Helen M. Byrne[2], Mary R. Myerscough[4] and David R. Greaves[1]

[1]Sir William Dunn School of Pathology, [2]Mathematical Institute, and [3]Department of Zoology, University of Oxford, Oxford, UK
[4]School of Mathematics and Statistics, University of Sydney, Sydney, Australia

HZF, 0000-0002-0457-7224; GSDP, 0000-0003-4118-1568; AEZ, 0000-0003-1824-7653; JAB, 0000-0003-1475-2898; HMB, 0000-0003-1771-5910; MRM, 0000-0002-4993-765X; DRG, 0000-0003-2856-9410

In both cells and animals, cannibalism can transfer harmful substances from the consumed to the consumer. Macrophages are immune cells that consume their own dead via a process called cannibalistic efferocytosis. Macrophages that contain harmful substances are found at sites of chronic inflammation, yet the role of cannibalism in this context remains unexplored. Here we take mathematical and experimental approaches to study the relationship between cannibalistic efferocytosis and substance accumulation in macrophages. Through mathematical modelling, we deduce that substances which transfer between individuals through cannibalism will concentrate inside the population via a coalescence process. This prediction was confirmed for macrophage populations inside a closed system. We used image analysis of whole slide photomicrographs to measure both latex microbead and neutral lipid accumulation inside murine bone marrow-derived macrophages ($10^4$–$10^5$ cells) following their stimulation into an inflammatory state *ex vivo*. While the total number of phagocytosed beads remained constant, cell death reduced cell numbers and efferocytosis concentrated the beads among the surviving macrophages. As lipids are also conserved during efferocytosis, these cells accumulated lipid derived from the membranes of dead and consumed macrophages (becoming macrophage foam cells). Consequently, enhanced macrophage cell death increased the rate and extent of foam cell formation. Our results demonstrate that cannibalistic efferocytosis perpetuates exogenous (e.g. beads) and endogenous (e.g. lipids) substance accumulation inside macrophage populations. As such, cannibalism has similar detrimental consequences in both cells and animals.

## 1. Introduction

Tissue homeostasis and inflammation resolution require macrophages to phagocytose pathogens and apoptotic cells (efferocytosis), including their own apoptotic macrophages (cannibalistic efferocytosis) [1–3]. Tissues that accumulate harmful stimuli (e.g. pathogens and necrotic cells) become inflamed and populated by large numbers of macrophages and other immune cells. Macrophage numbers increase via the recruitment and differentiation of monocytes from the bloodstream, and the proliferation of tissue-resident or monocyte-derived macrophages; they decrease via apoptosis (primarily) and emigration from the tissue [4–6]. Macrophages regulate inflammation via cytokine signalling and phagocytosis. These processes are primarily mediated by cytoplasmic or cell-surface pattern recognition receptors (PRRs) that detect pathogen- and damage-associated molecular patterns (PAMP/DAMPs) of pathogens and damaged cells. In the presence of PAMP/DAMPs and cytokines, macrophages polarize into a spectrum of pro- and anti-inflammatory states (e.g. M1 and M2) and produce cytokines that

orchestrate inflammation amplification and resolution [7]. The persistence of PAMP/DAMPs in the tissue or inside macrophages can cause chronic inflammation [8–10]. Consequently, the accumulation of pathogens and sterile substances inside macrophages is a hallmark of a variety of inflammatory diseases [11–14]; for example, *Mycobacterium tuberculosis* infections [15], neutral lipids and cholesterol crystals during atherosclerosis [11,16], monosodium urate and calcium pyrophosphate dihydrate crystals during gout and pseudogout [12], amyloid-β during Alzheimer's disease [13] and silica and asbestos during inflammation of the lung [14].

Inflammatory macrophages are short-lived phagocytes and, as such, gain substances (e.g. pathogens) from the dead macrophages which they consume, i.e. via cannibalistic efferocytosis [3,15,17,18]. Cannibalistic efferocytosis has been overlooked as a mechanism of substance accumulation in macrophages. Cannibalism, as seen in tumours, can be a beneficial mechanism that allows organisms to scavenge nutrients when the supply is low [19,20]. However, as seen in ecosystems, cannibalism can transfer harmful substances between individuals and perpetuate disease transmission (e.g. Kuru neurodegenerative disorder and bovine spongiform encephalopathy) [21]. Furthermore, indigestible substances that transfer from animal-to-animal via predation can accumulate to toxic levels along food chains (a process called biomagnification) [22]. Although conceivable, it is unknown if cannibalism and biomagnification have similar pathological consequences in macrophages as in animals. That is, do substances that transfer from cell-to-cell via cannibalistic efferocytosis (e.g. cholesterol and intracellular pathogens [15,17,18]) accumulate/biomagnify to potentially harmful levels inside macrophages?

In this study, we use experimental and mathematical approaches to elucidate the relationship between cannibalitic efferocytosis and substance accumulation in macrophages. We observe that efferocytosis fuses the contents of two macrophages into one, whereas division splits the contents of one macrophage between two. We derive a mathematical model (a coalescence process [23,24]) based on these observations. We then experimentally verify predictions from the mathematical model. Specifically, our experiments show that efferocytosis coalesces exogenous beads (derived from phagocytosis) and endogenous lipid (derived from the membranes of dead and consumed macrophages) inside murine macrophages following their stimulation into an inflammatory state *ex vivo*. The total quantity of indigestible substances contained inside macrophages is conserved during cannibalistic efferocytosis and thus concentrates inside the population as it decays in size via cell death. Furthermore, cannibalitic efferocytosis produces remarkable cell-to-cell variability [25]. Consequently, stimulation of cell death increased the rate, extent and variability of lipid accumulation in macrophages. Collectively, our results suggest that indigestible substances essentially biomagnify within inflammatory macrophage populations.

## 2. Results

### (a) Theoretical considerations

#### (i) Efferocytosis concentrates substances inside macrophage populations via a coalescence process

To better understand how substances redistribute among macrophages, we tracked the fate of intracellular beads as macrophages underwent apoptosis, efferocytosis and division (see the electronic supplementary material, methods). The time-lapse experiment of figure 1a shows that beads are conserved during apoptosis and are gained by the consumer cell during efferocytosis. Conversely, figure 1b shows that beads contained inside one cell are split between daughter cells during cell division.

These observations cast apoptosis/efferocytosis as a coalescence (or coagulation or aggregation) process [23] that fuses the cargo of two cells into one, and division as a branching (or fragmentation) process that splits the cargo of one cell between two (as illustrated in figure 1c). As such, bead redistribution within macrophages via apoptosis, efferocytosis and division may be viewed as a coagulation–fragmentation process [24]. Figure 1d depicts a historic lineage of apoptosis/efferocytosis (binary tree) which illustrates how beads coalesce from several cells with small numbers of beads per cell to one cell with large numbers of beads. When reversed, this binary tree describes how division dilutes bead numbers inside macrophages via a fragmentation process.

Bead conservation during apoptosis, efferocytosis and division implies that in a closed system the total number of beads within macrophage populations remains constant as the population changes in size. Consequently, the average number of beads per macrophage is inversely proportional to the total number of macrophages, i.e. halving as cell numbers double.

#### (ii) A mathematical model of substance accumulation as a coagulation–fragmentation process

To test hypotheses and to facilitate the interpretation of experimental results, we derived and analysed a simple coagulation–fragmentation model for bead accumulation inside macrophages. We strived to develop the simplest model which is consistent with experimental observations. The model distinguishes between live and apoptotic macrophages and keeps track of the number of beads that they contain. We consider macrophage apoptosis, efferocytosis and division with mass action kinetics displayed in figure 2a. We assume that these processes conserve, and are independent of, bead numbers. Furthermore, we assume dead cells are efferocytosed whole and neglect other forms of cell death (e.g. necrosis). These assumptions produce the following coagulation–fragmentation model:

$$\frac{\mathrm{d}}{\mathrm{d}t}\phi_n = \eta \underbrace{\sum_{n'=0}^{n}\phi_{n'}^{\dagger}\phi_{n-n'}}_{\text{efferocytosis source}} \underbrace{-\eta\phi_n\sum_{n'=0}^{\infty}\phi_{n'}^{\dagger}}_{\text{efferocytosis sink}} \underbrace{-\beta\phi_n}_{\text{apoptosis sink}}$$
$$+\underbrace{2\alpha\sum_{n'=n}^{\infty}\binom{n'}{n}\frac{\phi_{n'}}{2^{n'}}}_{\text{division source}} \underbrace{-\alpha\phi_n}_{\text{division sink}} \qquad (2.1)$$

and

$$\frac{\mathrm{d}}{\mathrm{d}t}\phi_n^{\dagger} = \underbrace{-\eta\phi_n^{\dagger}\sum_{n'=0}^{\infty}\phi_{n'}}_{\text{efferocytosis sink}} \underbrace{+\beta\phi_n}_{\text{apoptosis source}}, \qquad (2.2)$$

where $\phi_n = \phi_n(t)$ and $\phi_n^{\dagger} = \phi_n^{\dagger}(t)$ represent the number density of live and apoptotic macrophages that contain $n \geq 0$ beads at time $t \geq 0$, see the electronic supplementary material, methods for details. Overall, this equation is novel but shares common characteristics (e.g. convolution term) with other coagulation–fragmentation equations [24]. Note

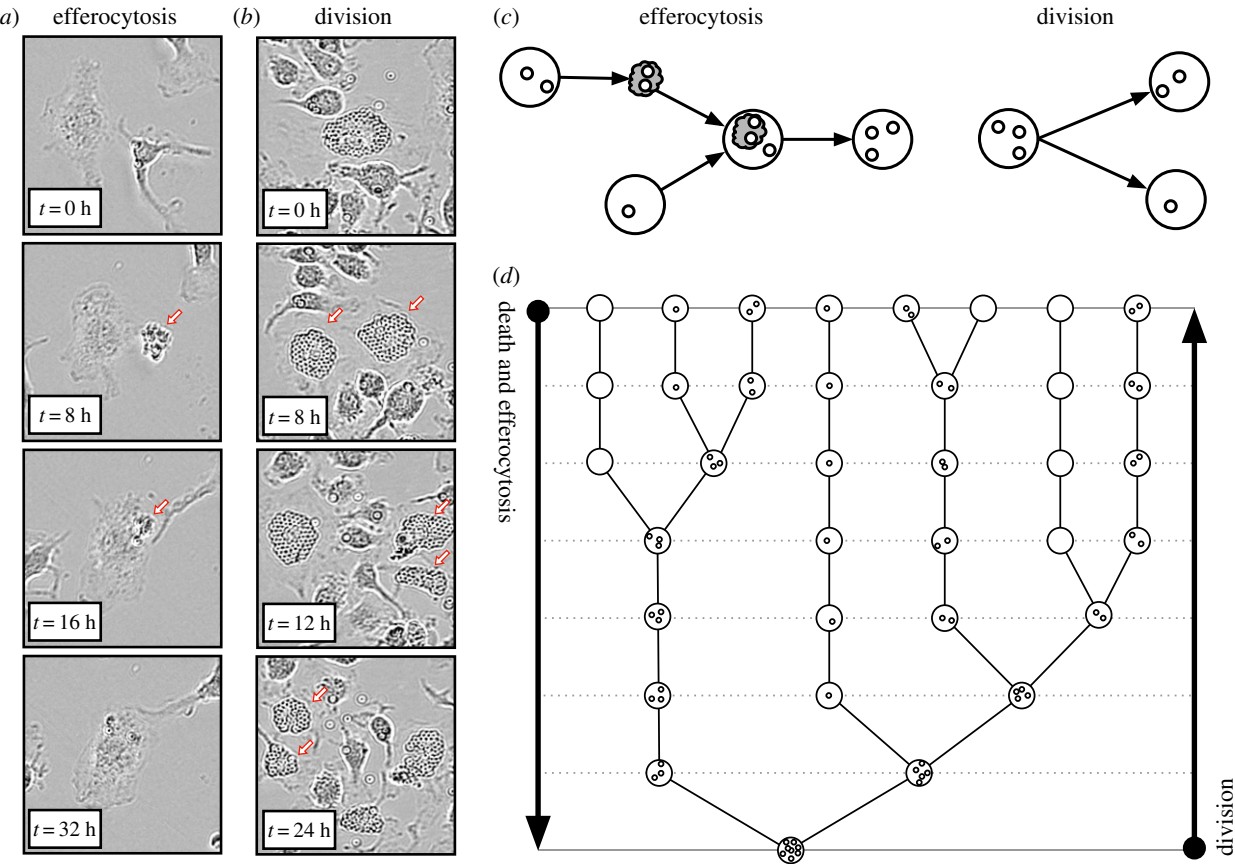

**Figure 1.** Efferocytosis fuses the contents of two cells into one. (*a*) Time-lapse microscopy of macrophage apoptosis (during times 0 and 8 h) and efferocytosis (during 16 and 32 h). The dead macrophage is indicated with an arrow. Intracellular beads are conserved during apoptosis and transferred from the consumed apoptotic cell to a live consumer cell during efferocytosis. (*b*) Time-lapse microscopy of macrophage division of one cell (during 0 and 8 h) and of both daughter cells (during 8 and 12 h, and 12 and 24 h). The daughter macrophages are indicated with arrows. Intracellular beads are split between daughter cells during division. (*c*) A schematic which illustrates (i) that apoptosis produces one apoptotic cell (grey cloud) from one live cell (white circle) with equal bead content (smaller circles), (ii) efferocytosis transfers the bead content of an apoptotic (consumed) cell to a live (consumer) cell, and (iii) division produces two daughter cells whose combined bead content is equal to the bead content of the parent cell. (*d*) A binary tree that represents how apoptosis/efferocytosis decreases the number of cells and increases the number of beads per cell by fusing the bead content of two live cells into one. In reverse, the same binary tree represents how cell division increases the number of cells and decreases the number of beads per cell by splitting the bead content of one cell between two. (Online version in colour.)

that this model describes accumulation of other substances such as lipids, sterile particles or pathogens.

Model simulation results for two cases are presented in figure 2*b,c*. These cases depict the theoretical distribution of beads among macrophages either without cell division ($\alpha = 0$, $\beta > 0$) or with cell division ($\alpha = \beta$, $\beta > 0$). Here, we assume that every cell initially contains one bead ($\phi_1(0) = N_0$ and $\phi_n(0) = 0$ for $n \neq 1$) and apoptotic cells are consumed instantaneously ($\eta \to \infty$).

Without cell proliferation, the model predicts that beads progressively and heterogeneously accumulate inside the population via efferocytosis as it decays exponentially in size via apoptosis. Figure 2*b* shows how beads are distributed across the population at different points in time in units of the average macrophage lifespan $\beta^{-1}$ (the key parameter in this case). As cell numbers decrease, this distribution becomes increasingly more uniform across a growing range of bead numbers per cell (typical of coalescence processes) [24]. That is, we expect the maximum and variability in number of beads per cell to increase as cells die. The faster cells die, the faster the maximum and variability grow.

If cells initially contain one bead and one unit quantity of endogenous substances (i.e. neutral lipid inside cell membranes), then the number of beads per cell at later time points might also represent the fold increase in the quantity of

endogenous substances (which are not degraded or removed) that macrophages accumulate via efferocytosis. For example, a macrophage with $n$ beads might also represent a macrophage that accumulates lipid derived from $n - 1$ other macrophages which have died and then have been consumed.

The model predicts that if cell division and apoptosis rates balance, then cell numbers remain constant and the population evolves to an equilibrium where the number of beads per cell remains small. Figure 2*c* shows how beads are distributed across the population at different points in time. The population tends to an equilibrium because division and apoptosis balance bead dilution/fragmentation and concentration/coagulation effects.

## (b) Experimental results
### (i) Bead accumulation inside macrophages
We designed an experiment to test the predictions of our mathematical model. We quantified latex bead (3 μm diameter) accumulation within primary murine bone marrow-derived macrophages (BMDMs) *ex vivo*. Differentiated BMDMs were incubated at a ratio of one latex bead (half red, half blue) per cell and then stimulated with interferon γ (IFNγ) and lipopolysaccharide (LPS). Unstimulated bead-loaded BMDMs were used as a control. Macrophages

**4**

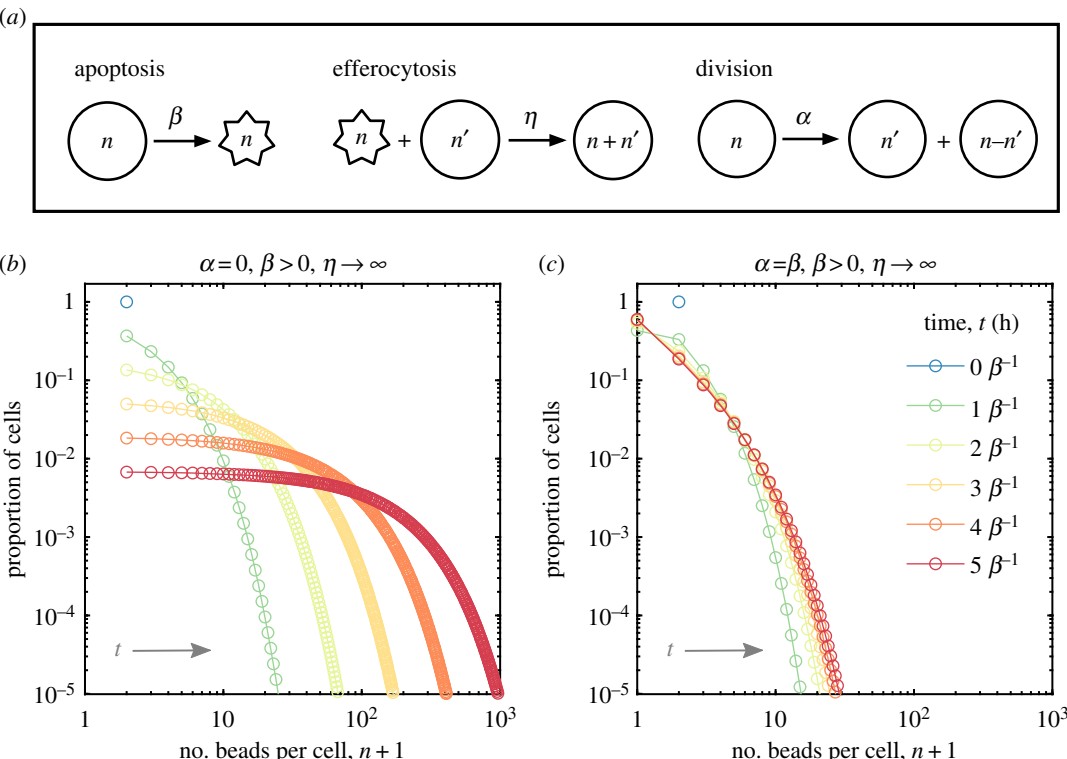

**Figure 2.** Coagulation–fragmentation model of bead accumulation in macrophages. (a) Cartoon schematic of the mass action kinetics for live (circles) and apoptotic (stars) macrophages due to apoptosis (rate $\beta$), efferocytosis (rate $\eta$) and cell division (rate $\alpha$). (b) Model prediction for the time-evolution of the proportion of cells with $n$ beads per cell while there is no division ($\alpha = 0$). In this case, apoptotic cells are instantly consumed ($\eta \to \infty$) and every cell initially ($N_0$ cells) contains $n = 1$ bead per cell. The population size $N$ declines as $N(t) = N_0 e^{-\beta t}$, where $t$ represents time (in hours). (c) Same as (b) except that the rates of apoptosis and division balance ($\alpha = \beta$) such that the population size remains constant over time ($N(t) = N_0$ cells). In both (b) and (c), each coloured line represents the predicted distribution after $t = 0, 1, \ldots, 5$ macrophage lifespans, $\beta^{-1}$ hours (from blue to red). (Online version in colour.)

stimulated with IFN$\gamma$ and LPS polarize to a classically activated (M1) state found early in acute inflammation and throughout chronic inflammation [7]. The number of beads per cell for every cell ($10^5$ cells and $10^5$ beads initially) was counted using a new in-house computer image recognition algorithm applied to whole slide photomicrographs (see the electronic supplementary material, methods).

The experiments presented in figure 3 show that efferocytosis concentrates beads inside macrophages following bead phagocytosis. Figure 3a shows images of bead-loaded BMDMs prior to stimulation (at time 0) and 24 and 48 h after stimulation with LPS and IFN$\gamma$. These images illustrate how both the magnitude of, and variation in, the number beads per cell increases as the number of BMDMs decreases. Also shown is an image of unstimulated BMDMs at 48 h. Unstimulated BMDMs proliferate to high confluency with low numbers of beads per cell. Figure 3b shows that the total number of internalized beads remained constant as the cell population size decreases following stimulation. This is consistent with observation from the pilot experiments shown in figure 1. We estimate the average death rate of bead-loaded BMDMs stimulated with LPS and IFN$\gamma$ to be $1/60\,\mathrm{h}^{-1}$ (see the electronic supplementary material). Figure 3c shows that the average number of beads per cell approximately doubled after each day from the initial state of roughly 1 bead per cell. Figure 3d shows how the beads were distributed across the BMDM population 0, 24 and 48 h after LPS and IFN$\gamma$ stimulation.

The experimental data are consistent with output from the mathematical model (see the electronic supplementary

material, methods) when (i) the initial condition of the model, i.e. the distribution of beads across the population, was equal to the experimental data, (ii) bead-loaded BMDMS stimulated with LPS and IFN$\gamma$ die with an average rate of $1/60\,\mathrm{h}^{-1}$, (iii) there is no cell division, and (iv) apoptotic cells are consumed instantaneously. Using the method of least squares, we find that mathematical model prediction fits the average values of the experimental data at 24 h with R-squared value $R^2 = 0.9929$ and at 48 h with R-squared value $R^2 = 0.9970$.

### (ii) Lipid accumulation inside M1 macrophages

The accumulation of lipid within macrophages (i.e. the generation of foam cells) is a feature of acute and chronic inflammatory responses, most notably in atherosclerotic plaques and tuberculosis granulomas [16,26]. Our mathematical model predicts that efferocytosis causes inflammatory macrophages to accumulate neutral lipid derived from the membranes of dead macrophages that have been consumed by other macrophages. To test this hypothesis experimentally, we use our image recognition algorithm to quantify oil red O-stained lipid droplets inside murine BMDMs stimulated with LPS and IFN$\gamma$ *ex vivo*. Unstimulated BMDMs were used as a control. Furthermore, we stimulated BMDMs with LPS and staurosporin (STPN), a compound that promotes apoptosis.

The experiments presented in figure 4 show that neutral lipids accumulate inside BMDMs in the same way that beads accumulate (figure 2). The images presented in figure 4a show how the quantity of accumulated lipid per cell increases as the number of cells decreases. For times up until 60 h, we

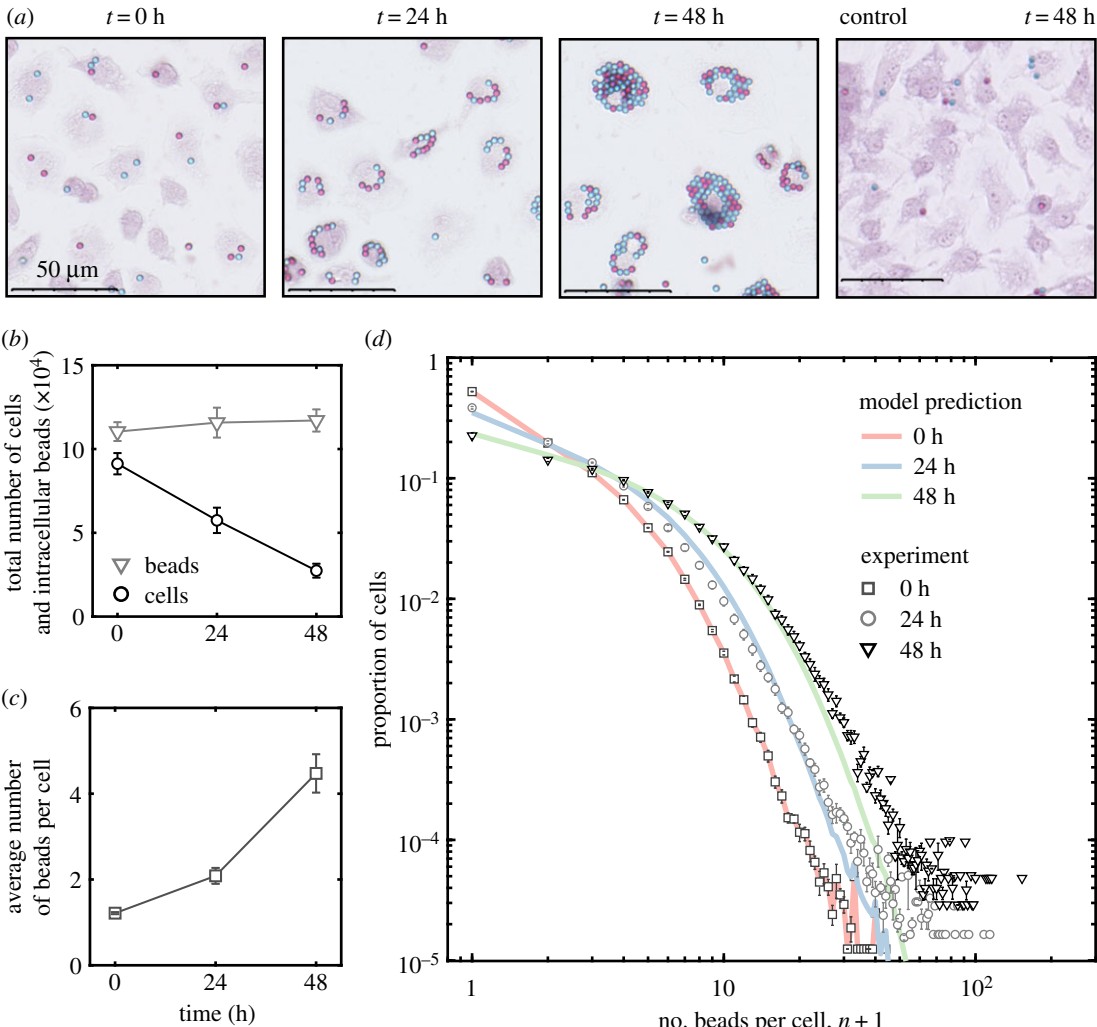

**Figure 3.** Bead accumulation in M1 macrophages. Quantification of red and blue beads inside safranin-stained murine bone marrow-derived macrophages (BMDMs) following stimulation with IFNγ and LPS. The data presented are averaged from four experiments carried with BMDMs derived from different mice on different days. Initially, there were approximately $9 \times 10^4$ cells that collectively contain $11 \times 10^4$ beads. (a) Representative images of BMDMs at 0, 24 and 48 h after stimulation. Also shown is a representative image of unstimulated BMDMs at 48 h (far right). (b) The total number of BMDMs (black circles) and the total number of intracellular beads (grey triangles) over time. (c) The average number of beads per cell over time. (d) The proportion of BMDMs with number $n$ beads per cell prior to stimulation (dark grey squares) and 24 (light grey circles) and 48 (black triangles) hours after stimulation. Also shown are predictions from the mathematical model (no cell division) run from the experimental initial condition (red line and black squares) with death rate $1/60$ $\mathrm{h}^{-1}$ to 24 (blue line) and 48 (green line) hours; the distribution have R-squared values $R^2 = 0.9929$ and $R^2 = 0.9970$ with the experimental distributions at 24 h at 48 h, respectively. (Online version in colour.)

observe that most apoptotic BMDMs, some of which contain accumulated lipid, are in the process of efferocytosis; most cells were dead after 60 h. Also shown is an image of unstimulated BMDMs and BMDMs stimulated with LPS and STPN at 48 h. Unstimulated BMDMs proliferate to high confluency without lipid accumulation. BMDMs stimulated with LPS and STPN accumulate more lipid than BMDMs stimulated with LPS and IFNγ. Figure 4b shows that the number of BMDMs declines from approximately $10^5$ (0 h) to $2 \times 10^4$ (60 h) after LPS and IFNγ stimulation, at which time almost all BMDMs stain positive for oil red O. Figure 4c shows that the average lipid content per cell grows exponentially with time, similar to bead accumulation shown in figure 3c and consistent with the predictions of our model. If neutral lipid accumulation were due to lipid synthesis or uptake (say, from low-density lipoprotein phagocytosis), then we would expect that the average lipid content per cell would either constantly increase or saturate with time. Figure 4d shows how the distribution of lipid across the population evolves with time. The extent of lipid accumulation per cell increases over time,

i.e. we see cells with larger amounts of lipid; and the cell-to-cell variation with respect to lipid content increases with time, i.e. we see larger disparities in lipid content between cells.

In figure 4, the quantity of intracellular lipid is expressed in arbitrary units (the cell area that stains positively for oil red O per cell). To facilitate model comparison, these arbitrary units should be expressed in terms of the average endogenous lipid content per macrophage (although this is likely to vary from cell to cell).

Like beads, neutral lipids are not degraded during apoptosis and efferocytosis and are unlikely to be degraded by, or removed from, M1 macrophages [16]. As such, the total quantity of neutral lipids (either as components of cellular membranes or lipid droplets) inside the macrophage population should remain constant over time. Efferocytosis essentially translocates the lipid inside the cell membranes (which are not stained by oil red O) of one cell into lipid droplets (which are stained by oil red O) of another cell, as illustrated in figure 5a. Consequently, the average quantity of accumulated lipid per cell increases as the population size

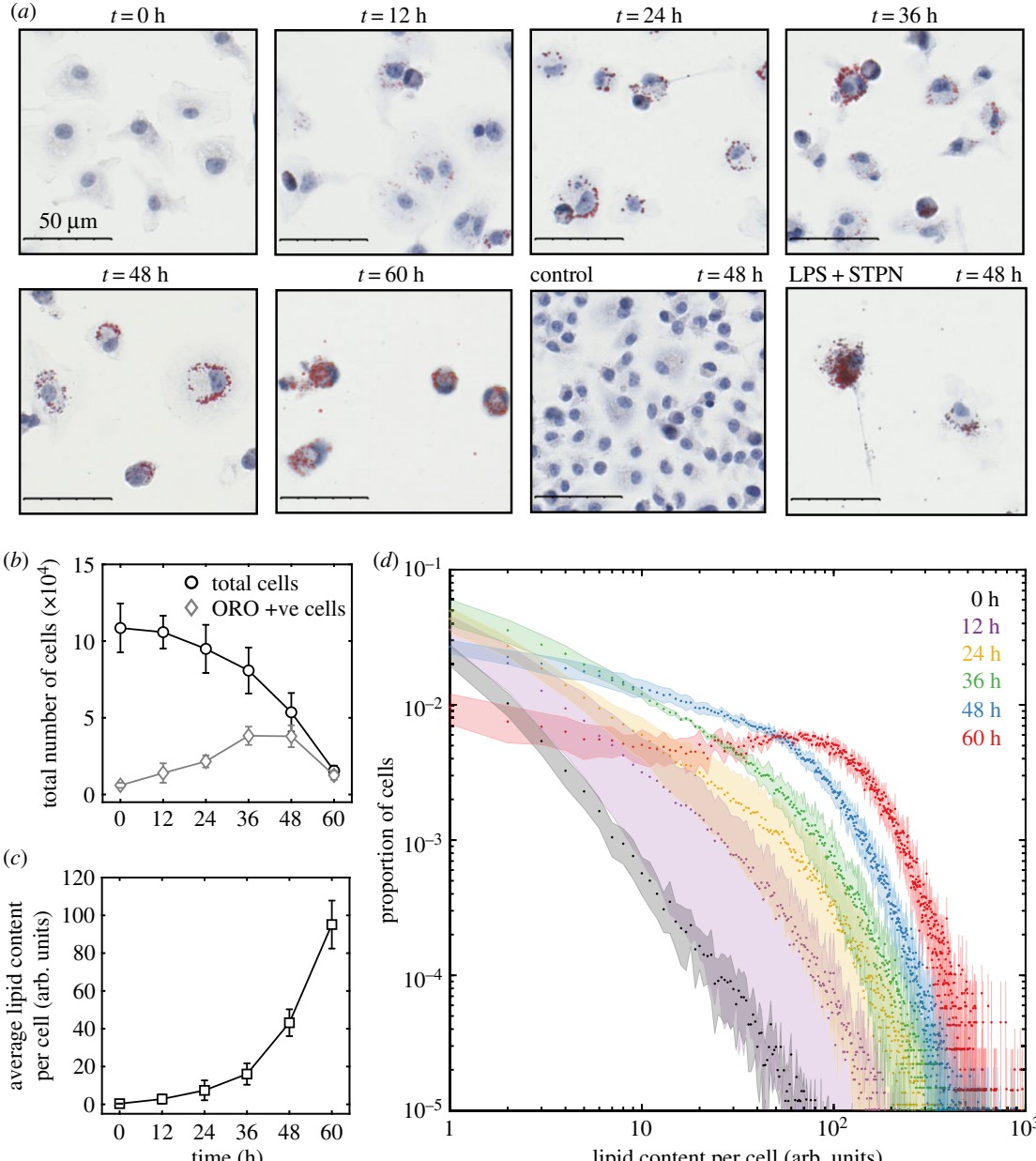

**Figure 4.** Neutral lipid accumulation in M1 macrophages. Quantification of oil red O-stained neutral lipids inside haematoxylin-stained murine bone marrow-derived macrophages (BMDMs) following stimulation with LPS and IFNγ. Initially, there were approximately $1.1 \times 10^5$ BMDMs with insignificant quantities of accumulated lipid. The data presented are averaged from four experiments carried out with BMDMs derived from different mice on different days. (*a*) Representative images of BMDMs at 0, 12, 24, 36, 48 and 60 h after stimulation. Also shown are images of unstimulated BMDMs at 48 h and BMDMs 48 h after stimulation with LPS and staurosporin (STPN). (*b*) The total number of BMDMs (black circles) and the number of BMDMs that stain positively for oil red O (ORO) (grey diamonds) over time. (*c*) The average lipid content per cell (arbitrary units) over time. (*d*) The proportion of BMDMs with a range of lipid contents (arbitrary units) at 0 (black), 12 (purple), 24 (yellow), 36 (green), 48 (blue) and 60 (red) hours after stimulation with LPS and IFNγ. (Online version in colour.)

decreases (figure 4*c*). This conservation property enables us to scale the arbitrary units of lipids in terms of the average quantity of lipid contained inside macrophage membranes.

Figure 5*b* shows that in BMDMs stimulated with either LPS and IFNγ, or LPS and STPN, the amount of accumulated lipid (inside lipid droplets) inside the whole population increases as endogenous lipid (inside cell membranes) decreases such that the total amount of lipid (endogenous+accumulated) remains constant over time. That is, the rate at which endogenous lipid is removed from the population via cell death is equal to the rate at which accumulated lipid is added to the population via efferocytosis. Furthermore, BMDMs stimulated with LPS and STPN die at a faster rate than those stimulated with LPS and IFNγ. We estimate the average death rate of

BMDMs stimulated with LPS and IFNγ to be $1/80\,\mathrm{h}^{-1}$ and with LPS and STPN to be $1/40\,\mathrm{h}^{-1}$ (see the electronic supplementary material). Consequently, BMDMs stimulated with LPS and STPN display a greater extent of lipid accumulation per cell than those stimulated with LPS and IFNγ, as predicted by the mathematical model.

Figure 5*c* shows how lipid is distributed across the BMDM population at 48 h after stimulation with LPS and IFNγ, or with LPS and STPN. Both distributions are in good qualitative agreement with predictions from the mathematical model. In this modelling case, we assume (i) all cells initially contain one unit of endogenous lipid each (inside cell membranes), (ii) BMDMs stimulated with LPS and IFNγ die with an average rate of $1/80\,\mathrm{h}^{-1}$, (iii)

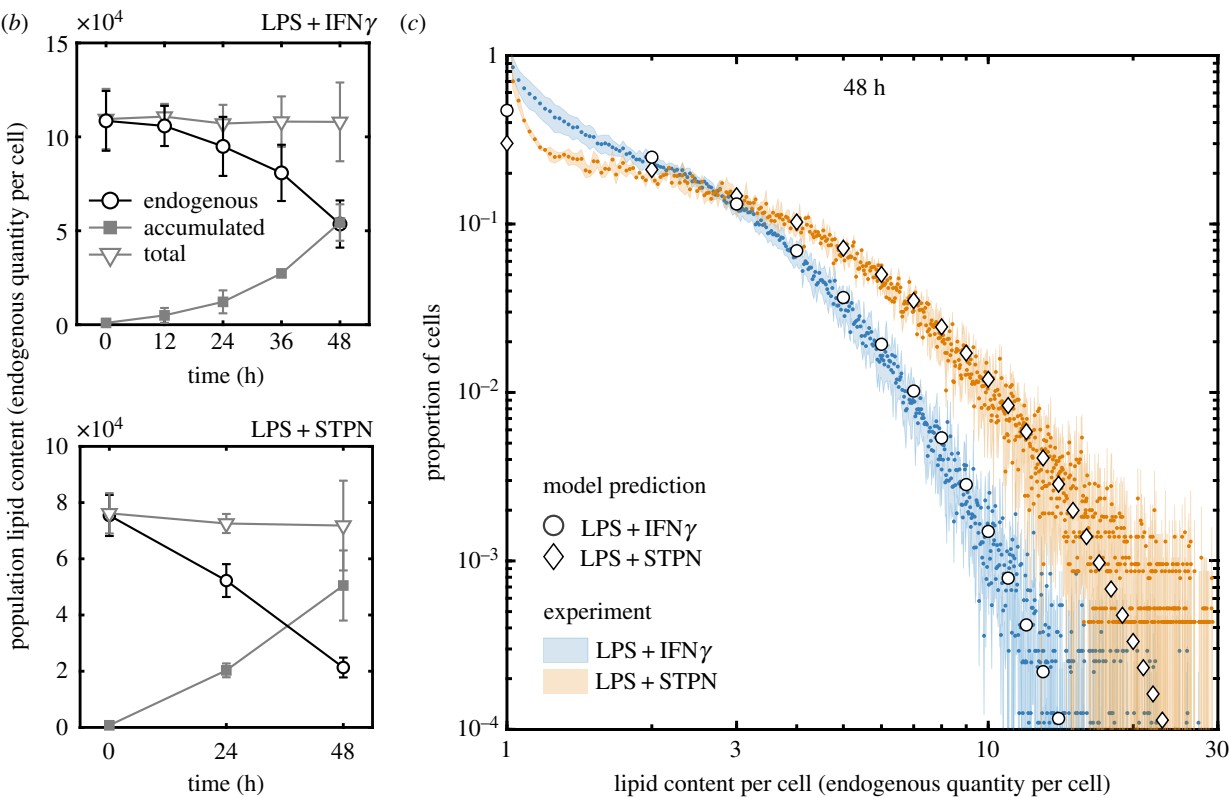

**Figure 5.** Enhanced apoptosis promotes foam cell formation inclusion of a dimension for lipid accumulation (in units of the endogenous quantity of lipid per cell) for model comparison. The data presented are averaged from four experiments (LPS + IFNγ) and three experiments (LPS + STPN) carried out with different mice on different days. (*a*) Cartoon illustration of lipid droplet accumulation (red) derived from the membranes of live cells (blue circle) that have since died (blue star) and have been consumed. Here two apoptosis and efferocytosis events are depicted to show the progression of lipid accumulation. (*b*) The quantity of endogenous lipid (black circles), the total quantity of accumulated lipid (filled grey squares) and their addition (light grey triangles) inside the whole BMDM population following stimulation with LPS and IFNγ (top) and LPS and STPN (bottom). (*c*) The proportion of cells with a range of lipid contents (in units of the quantity of endogenous lipid per cell) at 48 h after stimulation with LPS and IFNγ (blue) or LPS and STPN (orange). Also shown are predictions from the mathematical model (no cell division) run to 48 h from an initial condition where every cell contained 1 unit quantity of endogenous lipid with death rate $1/80\,\mathrm{h}^{-1}$ (circles) or $1/40\,\mathrm{h}^{-1}$ (diamonds). (Online version in colour.)

BMDMs stimulated with LPS and STPN die with an average rate of $1/40\,\mathrm{h}^{-1}$, (iv) cells do not divide, and (v) all apoptotic cells are consumed instantaneously (figure 2*d*).

## 3. Discussion

We hypothesized that cannibalistic efferocytosis contributes to the accumulation of substances inside inflammatory macrophages. A mathematical model (based on coagulation–fragmentation equations) was formulated and used to test this hypothesis. We concluded that the transfer of substances from dead to live macrophages via efferocytosis (figure 1) [15,18] continually concentrates these substances among surviving macrophages (figure 2). Model predictions were confirmed by quantifying latex microbead and neutral lipid accumulation inside murine bone marrow-derived macrophages following their stimulation into a pro-inflammatory state (M1) with LPS and IFNγ *ex vivo* (figures 3 and 5). A new in-house computer

vision algorithm was used to count the number of beads and quantify stained lipid droplets per cell.

Inflammatory macrophages which initially contained small quantities of phagocytosed beads and neutral lipid (in cell membranes) accumulated large quantities of beads and lipids (in lipid droplets) per cell (figures 3*a* and 4*a*). Neither beads nor neutral lipids are degraded during efferocytosis (figures 1, 3*b* and 5*a*) so that, as the population reduced in size, the average number of beads or lipid droplets per cell increased (figures 3*c* and 4*c*). Our mathematical model predicted how these substances are distributed among the surviving macrophages (figures 3*d*, 4*d* and 5*c*). That is, efferocytosis caused the indigestible contents of the macrophage population to concentrate and coalesce inside the macrophage population as it declined in size. Neutral lipids that accumulated derived from the membranes of macrophages that had died and then been consumed. Consequently, enhanced macrophage cell death (induced by staurosporin) increased the rate and extent of foam cell formation (figure 5).

Our model assumes that dead cells are consumed whole during early stages of apoptosis. This form of cell death appears to be dominant in our experiments. However, dead cells fragment into numerous particles during late stages of apoptosis and necrosis. If these forms of cell death were dominant (i.e. owing to inefficient efferocytosis), then the constituents of dead cells could be shared across multiple macrophages. This scenario will change how substances are distributed among macrophages and has been addressed in another modelling study (under review) [27].

Our findings suggest that any phagocytosed particle or pathogen that avoids subcellular degradation should perpetually accumulate within macrophages via efferocytosis. We experimentally demonstrated this for phagocytosed beads and endogenous neutral lipid. Consequently, we believe that this type of accumulation will occur in many other contexts. Our results suggest that efferocytosis is capable of driving the accumulation of phagocytosed particles inside macrophages during chronic inflammation. For example: (i) lung inflammation when alveolar macrophages accumulate airborne pollutants such as asbestos and silica crystals [14], (ii) brain inflammation (Alzheimer's disease) when microglia accumulate fibrilar amyloid-β [13], and (iii) artery wall inflammation (atherosclerosis) when monocyte-derived macrophages accumulate cholesterol crystals and neutral lipid [11]. Unlike in infections [18], cannibalistic efferocytosis has been overlooked as a mechanism of substance accumulation in sterile inflammation [9]. Numerous intracellular pathogens (e.g. *M. tuberculosis*) have evolved to avoid degradation and exploit efferocytosis by using dead and infected macrophages as a 'Trojan horse' to infect other macrophages [15,17,18]. We observe this effect for beads and lipids (figures 3 and 4) suggesting that it might also occur for sterile particles in general. That is, our results suggest that, together with pathogen replication and particle phagocytosis, the number of pathogens or sterile particles per cell can grow and diversify across macrophage populations via cannibalistic efferocytosis.

The accumulation of pathogens and sterile particles inside macrophages can induce cell death and can promote intracellular PRR activation and pro-inflammatory cytokine secretion (e.g. IL-1β) [8]. In this way, substance accumulation influences cell state such that macrophages might adopt similar characteristics to the macrophages which they consume. Consequently, a macrophage might become pro-inflammatory when it consumes a dead macrophage that contains a large quantity of pro-inflammatory substances. Alternatively, a macrophage might die when it consumes a dead macrophage with cytotoxic levels of pathogens or sterile particles. This 'serial killing' effect is seen in *M. tuberculosis*-infected macrophages [15] and might also be caused by other substances that promote cell death, such as cholesterol [16,28]. Because cell death and efferocytosis promote substance accumulation (figure 5) and substance accumulation can promote cell death [15,28], a detrimental positive feedback loop might arise which progressively increases the rate of cell death and substance accumulation inside macrophages [29].

Macrophage foam cells commonly appear in a variety of immune responses because PRR activation (e.g. by LPS) decreases cholesterol efflux from macrophages [16,26,30]. However, a decrease in lipid efflux from macrophages does not explain the source of lipid in macrophage foam cells. Our results suggest that foam cells arise in inflammatory macrophage populations because these cells slowly (relative to the death rate) remove accumulated lipid that is derived from the membranes of dead and consumed macrophages. Furthermore, foam cells secrete pro-inflammatory cytokines and are quick to die [28,31]. In this way, endogenous lipid could progressively accumulate within macrophage populations and cause macrophages to become increasingly pro-inflammatory as the immune response ages.

Similar to lipids, efferocytosis might also amplify the intracellular quantity of other endogenous substances, such as metabolites (e.g. uric acid) and trace elements (e.g. iron), which can also be pro-inflammatory and/or cytotoxic at high levels. For example, uric acid accumulation inside macrophages promotes inflammation associated with gout [12] and iron accumulation in macrophages promotes inflammation associated with venous leg ulcers [10].

In the light of our observations, we anticipate that macrophage division and emigration might be beneficial to the resolution of inflammation, whereas monocyte recruitment might be detrimental. Macrophage division reduces substance accumulation by splitting accumulated substances between daughter cells. As such, pro-inflammatory and harmful substances could dilute to benign levels within proliferating macrophages (figure 1) [5]. Furthermore, macrophage emigration could remove accumulated substances from the tissue, preventing it from recycling back into the local macrophage population via efferocytosis [6]. Lastly, monocyte recruitment introduces endogenous substances (e.g. neutral lipids) into the tissue which fuel their accumulation inside monocyte-derived macrophages via efferocytosis. We have extended the mathematical model presented here to model these type of effects on lipid accumulation inside macrophages during artery wall inflammation associated with atherosclerosis (under review) [27].

Our coalescence model generalizes beyond macrophages and draws parallels between cannibalistic cell and animal populations [20]. We have shown that cannibalism causes the average amount of substances per cell to exponentially increase with time (figures 3c and 4c). In ecosystems, the amount of inorganic substances per animal can increase exponentially along the trophic levels of a food chain (i.e. biomagnification) [22]. This biomagnification process arises from the transfer of indigestible substances via predation. Thus, biomagnification should also arise via cannibalism. In this way, substance accumulation via cannibalistic efferocytosis can be viewed as a biomagnification or coalescence process [23]. Intraspecific biomagnification (owing to cannibalism) could be particularly detrimental if it caused the accumulation of harmful substances. For example, either inorganic toxins or infectious agents could biomagnify to harmful levels inside single cannibalistic populations. This process could contribute to increased occurrences of diseases in animal populations where the causative agent propagates via cannibalism (e.g. prion diseases Kuru neurodegenerative disorder and bovine spongiform encephalopathy) [21].

This study contributes to a growing body of evidence that casts cannibalitic efferocytosis as a double-edged sword [15,18]. Although crucial for tissue homeostasis and inflammation resolution [2,3], efferocytosis also perpetuates subcellular substance accumulation which might contribute to the pathogenesis of inflammation. That is, impaired efferocytosis stunts substance accumulation via cannibalistic efferocytosis but also enhances the likelihood of postapoptotic necrosis [2]. In this light, cannibalistic efferocytosis is an underappreciated component of macrophage population dynamics with similar detrimental consequences to cannibalism in animal populations.

**Ethics.** Human serum was obtained from human volunteers who had given informed consent and with ethical approval from the appropriate local ethics committee. Animal studies were performed with local ethical approval from the Dunn School of Pathology Animal Welfare Ethical Review Board and according to the United Kingdom Home Office regulations (Guidance on the Operation of Animals, Scientific Procedures Act, 1986). C57BL/6 mice were obtained from the Biomedical Sciences Unit (Oxford, United Kingdom) and were housed in a 12 h light/dark cycle with free access to food and water.

**Data accessibility.** Data available from the Dryad Digital Repository: http://dx.doi.org/10.5061/dryad.c3269fc [32]. These data include the whole-slide photomicrographs of bead and lipid accumulation inside murine macrophage population stimulated with LPS and IFNγ.

**Authors' contributions.** H.Z.F.: conceptualization, data curation, formal analysis, investigation, methodology, software, validation, visualization, writing—original draft preparation, writing—review and editing. L.Z.: investigation, methodology, resources, supervision. G.S.D.P.: investigation, methodology, resources, supervision, writing—review and editing. A.t.B.: investigation, methodology, resources. A.E.Z.: formal analysis, validation, writing—review and editing. J.A.B.: software. H.M.B.: conceptualization, methodology, supervision, writing—review and editing. M.R.M.: conceptualization, funding acquisition, methodology, supervision, writing—review and editing. D.R.G.: conceptualization, funding acquisition, methodology, project administration, resources, supervision, writing—review and editing.

**Competing interests.** We declare we have no competing interests.

**Funding.** D.R.G. and H.Z.F. acknowledge support from the British Heart Foundation (programme grant RG/15/10/31485 to D.R.G.). M.R.M. and H.Z.F. acknowledge support from the Australian Research Council (Discovery Program grant no. DP160104685 to M.R.M.).

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
