## [Reviewer comments · Proceedings of the Royal Society B: Biological Sciences]

Review History

RSPB-2019-0730.R0 (Original submission)

Review form: Reviewer 1

Recommendation

Accept with minor revision (please list in comments)

Scientific importance: Is the manuscript an original and important contribution to its field?

Excellent

General interest: Is the paper of sufficient general interest?

Excellent

Quality of the paper: Is the overall quality of the paper suitable?

Good

Is the length of the paper justified?

Yes

Should the paper be seen by a specialist statistical reviewer?

No

Do you have any concerns about statistical analyses in this paper? If so, please specify them explicitly in your report.

No

It is a condition of publication that authors make their supporting data, code and materials available - either as supplementary material or hosted in an external repository. Please rate, if applicable, the supporting data on the following criteria.

Is it accessible?

N/A

Is it clear?

N/A

Is it adequate?

N/A

Do you have any ethical concerns with this paper?

No

Comments to the Author

This is a really nice paper using a mathematical modelling approach to examine the contribution of cell death and phagocytosis to the process of accumulation of material within macrophages. It is well written and relatively easy to follow. The close match between experimental data and that predicted by the model is of great interest. However, it was not clear what the impact of changing different parameters would have on the theoretical distributions shown. Perhaps that would be worthy of further discussion?

A couple of minor points.

1) The title specifically refers to "efferocytosis". I wondered how the authors could be certain that the mechanism proposed would be specific for phagocytosis of apoptotic cells? What about necrotic material? What would happen if apoptotic cells were opsonised with IgG and cleared by FcR-dependent pathways. Also there are different molecular mechanisms for apoptotic cell clearance. Does it make any difference which pathways are involved? These issues should be discussed.

2) The authors assume one cell generates a single apoptotic particle. What happens if macrophages form multiple apoptotic bodies, perhaps distributing beads randomly into the resultant cell fragments? Again this might be something worth considering in Discussion.

Review form: Reviewer 2

Recommendation

Accept with minor revision (please list in comments)

Scientific importance: Is the manuscript an original and important contribution to its field?

Good

General interest: Is the paper of sufficient general interest?

Good

Quality of the paper: Is the overall quality of the paper suitable?

Good

Is the length of the paper justified?

No

Should the paper be seen by a specialist statistical reviewer?

No

Do you have any concerns about statistical analyses in this paper? If so, please specify them explicitly in your report.

No

It is a condition of publication that authors make their supporting data, code and materials available - either as supplementary material or hosted in an external repository. Please rate, if applicable, the supporting data on the following criteria.

Is it accessible?

Yes

Is it clear?

Yes

Is it adequate?

N/A

Do you have any ethical concerns with this paper?

No

Comments to the Author

This is a thorough paper that combines experiments with modeling to track the distribution of beads and neutral lipids in populations of macrophages. On the whole, it is a careful and complete exposition.

I have only a few minor suggestions for improving the work.

The authors have written this paper with great attention to clarity. In places, there is repetition of some ideas. It seems that the paper could be reduced by 10-15% by avoiding such repetitions, and by referring some of the detailed methods (repeated in Figure captions) to the Appendix or Supplement. Similarly, using a few tables would be clearer than verbal text, e.g. in Lines 28-32 of Suppl 2, and other places where numerical information is summarized.

The intro to the paper could use some key focal questions that would be posed here and answered in the paper. Are there hypotheses to distinguish between? Is the work going to provide specific conclusions? At the moment, the Intro is descriptive.

It is not fully clear to this reviewer why the model was entirely relegated to Supplement 1. The results of the model are discussed before the model is produced, which is confusing. Possibly a basic model equation could be placed in the main text, with key definitions (in a table) while the model variants and detailed analysis would be kept in a supplement.

Coagulation/fragmentation models have a long history. It would be useful to have comments

that identify which parts of the model in this paper were already derived and solved by others for other coalescence/fragmentation situations, and which are entirely new here. Possibly this may reduce the need for level of detail.

The authors did not cite a paper of relevance to this work:

[1] Marée et al (2004) J Theor Biol 233(4): 533-551.

While there was no cell division or death, the idea of macrophages ingesting apoptotic cells and several models for that process is shared. A link to inflammation was made also, but based on a different mechanism. It makes sense to link to that work.

The model is, by and large, presented clearly

Definitions (Lines 6-12 in Suppl 1) are a bit wordy, and could be condensed and made easier to read in a Table.

In the model, the parameters for cell division, death, apoptosis, etc are assumed to be independent of cell state (number of particles). How accurate is this? it is a good starting assumption, and maybe a comment or two about its accuracy would be warranted. Fig 4 of Suppl 2 suggests that the death rate is not strictly constant.

It seems intuitively obvious that if cells are dying but number of beads (or lipid) is conserved, then there will be more beads (or lipid) per remaining cell. So it would be important to add other insights that the model(s) can provide beyond this. For example, in [1] the authors used several models, together with Akaike analysis to draw conclusions that were not a priori clear about what was the underlying mechanism for observed macrophage ingestion dynamics. Probably this was (or could be) true here too, but the paper was not written with this idea in mind.

How do authors quantify number of dead cells vs bits of apoptotic material? I was not clear on this point.

The following sentence is maybe the main point of the paper and is hidden in the Discussion: "cannibalistic efferocytosis has been overlooked as a mechanism of substance accumulation". So the Intro should possibly focus on this theme more clearly.

The authors state (line 266) that

" macrophage division and emigration might be beneficial to the resolution of inflammation.."

This sentence needs to be qualified, as it could be misinterpreted.

Typos:

Suppl 1 sec 1.7: Fix:

Substituting equations (27) and (??)

Line 64:

observed that macrophages
observed than macrophages

Line 78

macrophages numbers
macrophage numbers

Decision letter (RSPB-2019-0730.R0)

23-Apr-2019

Dear Mr Ford:

Your manuscript has now been peer reviewed and the reviews have been assessed by an Associate Editor. The reviewers' comments (not including confidential comments to the Editor) and the comments from the Associate Editor are included at the end of this email for your reference. As you will see, the reviewers are positive but have raised some issues that we would like you to address.

Research ethics:

Use of animals and field studies:

It is a condition of publication that you make available the data and research materials supporting the results in the article. Datasets should be deposited in an appropriate publicly available repository and details of the associated accession number, link or DOI to the datasets

must be included in the Data Accessibility section of the article (<https://royalsociety.org/journals/ethics-policies/data-sharing-mining/>). Reference(s) to dataset(s) should also be included in the reference list of the article with DOIs (where available).

Please submit a copy of your revised paper within three weeks. If we do not hear from you within this time your manuscript will be rejected. If you are unable to meet this deadline please let us know as soon as possible, as we may be able to grant a short extension.

Best wishes,
Proceedings B
<mailto:proceedingsb@royalsociety.org>

Associate Editor
Comments to Author:

The referees are in agreement that this is an interesting and useful paper that is a great example of an experimental/modelling collaboration. They also raise some important points that should be considered carefully in the next draft of the paper.

Reviewer(s)' Comments to Author:

Referee: 1

Comments to the Author(s)

This is a really nice paper using a mathematical modelling approach to examine the contribution of cell death and phagocytosis to the process of accumulation of material within macrophages. It is well written and relatively easy to follow. The close match between experimental data and that predicted by the model is of great interest. However, it was not clear what the impact of changing different parameters would have on the theoretical distributions shown. Perhaps that would be worthy of further discussion?

A couple of minor points.

1) The title specifically refers to "efferocytosis". I wondered how the authors could be certain that the mechanism proposed would be specific for phagocytosis of apoptotic cells? What about necrotic material? What would happen if apoptotic cells were opsonised with IgG and cleared by FcR-dependent pathways. Also there are different molecular mechanisms for apoptotic cell clearance. Does it make any difference which pathways are involved? These issues should be discussed.

2) The authors assume one cell generates a single apoptotic particle. What happens if macrophages form multiple apoptotic bodies, perhaps distributing beads randomly into the resultant cell fragments? Again this might be something worth considering in Discussion.

Referee: 2

Comments to the Author(s)

This is a thorough paper that combines experiments with modeling to track the distribution of beads and neutral lipids in populations of macrophages. On the whole, it is a careful and complete exposition.

I have only a few minor suggestions for improving the work.

The authors have written this paper with great attention to clarity. In places, there is repetition of some ideas. It seems that the paper could be reduced by 10-15% by avoiding such repetitions, and by referring some of the detailed methods (repeated in Figure captions) to the Appendix or Supplement. Similarly, using a few tables would be clearer than verbal text, e.g. in Lines 28-32 of Suppl 2, and other places where numerical information is summarized.

The intro to the paper could use some key focal questions that would be posed here and answered in the paper. Are there hypotheses to distinguish between? Is the work going to provide specific conclusions? At the moment, the Intro is descriptive.

It is not fully clear to this reviewer why the model was entirely relegated to Supplement 1. The results of the model are discussed before the model is produced, which is confusing. Possibly a basic model equation could be placed in the main text, with key definitions (in a table) while the model variants and detailed analysis would be kept in a supplement.

Coagulation/fragmentation models have a long history. It would be useful to have comments that identify which parts of the model in this paper were already derived and solved by others for other coalescence/fragmentation situations, and which are entirely new here. Possibly this may reduce the need for level of detail.

The authors did not cite a paper of relevance to this work:

[1] Marée et al (2004) J Theor Biol 233(4): 533-551.

While there was no cell division or death, the idea of macrophages ingesting apoptotic cells and several models for that process is shared. A link to inflammation was made also, but based on a different mechanism. It makes sense to link to that work.

The model is, by and large, presented clearly

Definitions (Lines 6-12 in Suppl 1) are a bit wordy, and could be condensed and made easier to read in a Table.

In the model, the parameters for cell division, death, apoptosis, etc are assumed to be independent of cell state (number of particles). How accurate is this? it is a good starting assumption, and maybe a comment or two about its accuracy would be warranted. Fig 4 of Suppl 2 suggests that the death rate is not strictly constant.

It seems intuitively obvious that if cells are dying but number of beads (or lipid) is conserved, then there will be more beads (or lipid) per remaining cell. So it would be important to add other insights that the model(s) can provide beyond this. For example, in [1] the authors used several models, together with Akaike analysis to draw conclusions that were not a priori clear about what was the underlying mechanism for observed macrophage ingestion dynamics. Probably this was (or could be) true here too, but the paper was not written with this idea in mind.

How do authors quantify number of dead cells vs bits of apoptotic material? I was not clear on this point.

The following sentence is maybe the main point of the paper and is hidden in the Discussion: "cannibalistic efferocytosis has been overlooked as a mechanism of substance accumulation". So the Intro should possibly focus on this theme more clearly.

The authors state (line 266) that

" macrophage division and emigration might be beneficial to the resolution of inflammation.."

This sentence needs to be qualified, as it could be misinterpreted.

Typos:

Suppl 1 sec 1.7: Fix:

Substituting equations (27) and (??)

Line 64:

observed that macrophages
observed than macrophages

Line 78

macrophages numbers
macrophage numbers

Author's Response to Decision Letter for (RSPB-2019-0730.R0)

See Appendix A.

Decision letter (RSPB-2019-0730.R1)

13-May-2019

Dear Mr Ford

I am pleased to inform you that your manuscript entitled "Efferocytosis perpetuates substance accumulation inside macrophage populations" has been accepted for publication in Proceedings B.

Open Access

You are invited to opt for Open Access, making your freely available to all as soon as it is ready for publication under a CC BY licence. Our article processing charge for Open Access is £1700. Corresponding authors from member institutions (<http://royalsocietypublishing.org/site/librarians/allmembers.xhtml>) receive a 25% discount to these charges. For more information please visit <http://royalsocietypublishing.org/open-access>.

Paper charges

Sincerely,

Proceedings B
mailto: proceedingsb@royalsociety.org

Associate Editor:

Board Member

Comments to Author:

Lines 28-30 and 290-297 please check spelling of biomagnification/biomagnification and make consistent.

Lines 36, 45 and 299 spelling of cannibalistic.

Appendix A

Response to Reviewers Comments

Our Manuscript number = RSPB-2019-0730

Research ethics:

We have added details of research ethics for human serum to the methods section of our revised manuscript (see lines 75-77 of the supplementary material).

Use of animals:

If your study uses animals please include details in the methods section of any approval and licences given to carry out the study and include full details of how animal welfare standards were ensured.

We have added details of research ethics for animal studies and animal welfare standards to the methods section of our revised manuscript (see lines 57-62 of the supplementary material).

If you wish to submit your data to Dryad (<http://datadryad.org/>) and have not already done so you can submit your data via this link [http://datadryad.org/submit?journalID=RSPB&manu=\(Document](http://datadryad.org/submit?journalID=RSPB&manu=(Document) not available), which will take you to your unique entry in the Dryad repository.

We have uploaded our whole slide pictomicrographs to Dryad and have added a data accessibility section (see lines 105-107 of the supplementary material).

We have amalgamated the original two supplementary pdf documents into one document as requested. The document also includes the methods section.

The online supplementary document has been updated as per your guidance.

Best wishes,

Proceedings B

Associate Editor

Comments to Author:

The referees are in agreement that this is an interesting and useful paper that is a great example of an experimental/modelling collaboration. They also raise some important points that should be considered carefully in the next draft of the paper.

Dear Associate Editor

Many thanks for sending us reviewers' comments and details of how to re-format the manuscript for publication in Proceedings B.

We were impressed by the quality and thoughtfulness of the reviewers' comments and the quick turn around time of the reviewing process.

We have provided a point-by-point response to the comments and suggestions of the two reviewers.

Yours faithfully,

Hugh Z Ford and David R greaves

Referee: 1

Comments to the Author(s)

This is a really nice paper using a mathematical modelling approach to examine the contribution of cell death and phagocytosis to the process of accumulation of material within macrophages.

It is well written and relatively easy to follow. The close match between experimental data and that predicted by the model is of great interest.

Thank you!

However, it was not clear what the impact of changing different parameters would have on the theoretical distributions shown. Perhaps that would be worthy of further discussion?

Following Reviewer 1's suggestions we have re-written sections of the revised manuscript to address how different parameters affect the theoretical distributions.

Time in our model is scaled by the macrophage death rate, the key parameter in our model. Thus the rate and extent of substance accumulation is proportional to the death rate. We have updated the main text to clarify this point (see lines 92-93 and 95-97 of the main text). For example, an enhanced death rate enhanced substance accumulation via efferocytosis. This prediction was verified experimentally by treating cells with staurosporin, which induces cell death (Figure 5). As predicted by our model, we observed an increase in both the extent and rate of lipid accumulation inside macrophages treated with staurosporin (derived from the membranes of dead and consumed cells).

The other two parameters in our model were the division and efferocytosis rate. A detailed investigation into the effects of altered division and efferocytosis rates is now included in the supplementary material (see Figure 6 and 7 in the supplementary material). Two different rates of division (zero or equal to the death rate) are shown in Figure 2 of the main text (see lines 86-88 of the main text). Pro-inflammatory M1 macrophages do not divide. Furthermore, we assumed that apoptotic cells were instantaneously consumed when comparing our model to experimental data. For relevant periods of time, our mathematical model produces theoretical distributions that are similar for biologically plausible values of efferocytosis and instant efferocytosis. Further details on altered efferocytosis rates is included in another manuscript (see <https://www.biorxiv.org/content/10.1101/557538v1>).

1) The title specifically refers to "efferocytosis". I wondered how the authors could be certain that the mechanism proposed would be specific for phagocytosis of apoptotic cells? What about necrotic material?

We thank Reviewer 1 for raising this important point as it warrants further discussion and investigation. It is possible that macrophages could also gain substances derived from the debris of necrotic macrophages. We have updated the manuscript to address this point (see lines 222-237 of the main text). Unfortunately, manuscript length constraints prevent us from addressing it in detail here. In our manuscript, we assumed that efficient macrophage efferocytosis prevented apoptotic necrosis in our study. Indeed we have started to develop mathematical models to include consideration of cell necrosis and necrotic cell debris in another manuscript (see <https://www.biorxiv.org/content/10.1101/557538v1>).

What would happen if apoptotic cells were opsonised with IgG and cleared by FcR-dependent pathways.

In the experiments reported in this manuscript we opsonised our microbeads using pooled human serum which contains antibodies but is unlikely to contain a high titre of antibodies that specifically recognise unconjugated latex beads. Reviewer 1's question would probably best be addressed by using classic macrophage phagocytosis protocols that use opsonised sheep red blood cells, however this would lack the ease and accuracy of automated cell and bead counting that we have developed and validated in this report.

Also there are different molecular mechanisms for apoptotic cell clearance. Does it make any difference which pathways are involved? These issues should be discussed.

There should be little difference in the experimental results if these different pathways resulted in swift dead cell clearance. However, we anticipate that defects in pathways which ultimately reduce the apoptotic cell clearance rate will reduce the substances accumulate rate via efferocytosis and increase the likelihood of postapoptotic necrosis (see lines 222-227 and 302-303 of the main text). This is an interesting point as necrotic cells fragment into particle which could be shared by multiple macrophages. Having developed methods for reliable quantitative measurement of macrophage phagocytosis in future studies we could use macrophages from genetically altered animals known to have specific defects in different pathways known to be important for the recognition of apoptotic cells.

2) The authors assume one cell generates a single apoptotic particle. What happens if macrophages form multiple apoptotic bodies, perhaps distributing

beads randomly into the resultant cell fragments? Again this might be something worth considering in Discussion.

It is true that there are several types of cell death and mechanisms for dead cell clearance. The sharing of dead cell constituents was considered as an alternative hypothesis. We have updated the manuscript to address this point (see lines 222-227 of the main text). However, we did not observe this behaviour in our experiments (see Figure 1) and the mathematical model generated by this assumption is not consistent with our experimental data. Specifically, this model failed to produce the large extent of substance accumulation and the large cell-to-cell variation which we see experimentally. We have updated the manuscript to clarify these point (see lines 21-26 of the supplementary material).

While it is possible to introduce more biological complexity into our model, we strived to create the simplest model that was consistent experimental observations. Indeed, we have already developed a more sophisticated mathematical model that includes consideration of postapoptotic necrosis and the uptake of necrotic cell debris <https://www.biorxiv.org/content/10.1101/557538v1>

Referee: 2

Comments to the Author(s)

This is a thorough paper that combines experiments with modeling to track the distribution of beads and neutral lipids in populations of macrophages. On the whole, it is a careful and complete exposition.

Thank you!

I have only a few minor suggestions for improving the work.

The authors have written this paper with great attention to clarity. In places, there is repetition of some ideas. It seems that the paper could be reduced by 10-15% by avoiding such repetitions, and by referring some of the detailed methods (repeated in Figure captions) to the Appendix or Supplement. Similarly, using a few tables would be clearer than verbal text, e.g. in Lines 28-32 of Suppl 2, and other places where numerical information is summarized.

In revising our manuscript we have tried to retain the clarity of the original version while avoiding excessive repetition and respecting the word limit of this journal (see legend of Figure 2 for example).

The intro to the paper could use some key focal questions that would be posed here and answered in the paper. Are there hypotheses to distinguish

between? Is the work going to provide specific conclusions? At the moment, the Intro is descriptive.

This is a good point and we have revised the introduction to set out specific hypotheses, research questions and conclusions (see lines 29-33 of the main text). As discussed with Reviewer 1, the sharing of dead cell constituents was considered as an alternative hypothesis. We have updated the manuscript to address this point (see lines 222-227 of the main text). However, we did not observe this behaviour in our experiments (see Figure 1) and the mathematical model generated by this assumption is not consistent with our experimental data. Specifically, this model failed to produce the large extent of substance accumulation and the large cell-to-cell variation which we see experimentally. We have updated the manuscript to clarify these point (see lines 21-26 of the supplementary material).

It is not fully clear to this reviewer why the model was entirely relegated to Supplement 1. The results of the model are discussed before the model is produced, which is confusing. Possibly a basic model equation could be placed in the main text, with key definitions (in a table) while the model variants and detailed analysis would be kept in a supplement.

We agree with this comment and we have now moved the main equations (equations (11) & (12) in supplementary paper 1) into the results section of the main text of the manuscript to accompany the model output presented in Figure 2 (see lines 80-81 of the main text).

Coagulation/fragmentation models have a long history. It would be useful to have comments that identify which parts of the model in this paper were already derived and solved by others for other coalescence/fragmentation situations, and which are entirely new here. Possibly this may reduce the need for level of detail.

Reviewer 2 is correct, coagulation-fragmentation models have a long history (>100 years) and deserve further discussion. Regarding the novelty of the equations, we have introduced one sentence in the main text (see lines 83-83) and several discussions in the supplementary file (lines 207-208 and 263-266). Unfortunately, the word limit of the journal prevented a full discussion of this area of mathematics in the main text.

The authors did not cite a paper of relevance to this work:

[1] Marée et al (2004) J Theor Biol 233(4): 533-551.

While there was no cell division or death, the idea of macrophages ingesting apoptotic cells and several models for that process is shared. A link to inflam-

mation was made also, but based on a different mechanism. It makes sense to link to that work.

We thank Reviewer 2 for bringing this highly relevant study to our attention. We have cited this paper in the revised version of our manuscript (see lines 191-193 of the supplementary text).

The model is, by and large, presented clearly. Definitions (Lines 6-12 in Suppl 1) are a bit wordy, and could be condensed and made easier to read in a Table.

We have revised the text to clarify our definitions (see lines 167-168 of the supplementary text).

In the model, the parameters for cell division, death, apoptosis, etc are assumed to be independent of cell state (number of particles). How accurate is this? it is a good starting assumption, and maybe a comment or two about its accuracy would be warranted. Fig 4 of Suppl 2 suggests that the death rate is not strictly constant.

We have considered this and have revised the text accordingly. It has been shown that the macrophage death rate is an increasing function with respect to the cellular lipid content. Although this line of investigation was beyond the scope of this study, we have elaborated on this point in the supplementary material (see lines 154-161) and have briefly discussed the consequences of this (see lines 255-257 of the main text).

While it is possible to introduce this level of detail into our model, we strived to create the simplest model that was consistent experimental observations. The agreement between our simple model and the experimental data did not warrant the development of more sophisticated models.

We have introduced more biological complexity into our model to study the dynamics of macrophage lipid accumulation inside atherosclerotic plaques (see <https://www.biorxiv.org/content/10.1101/557538v1>). A current project of ours builds upon this model to investigate the consequences of a macrophage death rate that increases with cellular lipid content.

It seems intuitively obvious that if cells are dying but number of beads (or lipid) is conserved, then there will be more beads (or lipid) per remaining cell. So it would be important to add other insights that the model(s) can provide beyond this.

We thank Reviewer 2 for this comment. We have updated the main text to highlight how cell-to-cell variation arises via cannibalistic efferocytosis, which is perhaps more counterintuitive (see lines 45 and 95-97 of the main text).

For example, in [1] the authors used several models, together with Akaike analysis to draw conclusions that were not a priori clear about what was the underlying mechanism for observed macrophage ingestion dynamics. Probably this was (or could be) true here too, but the paper was not written with this idea in mind.

The assumptions of our model were informed by observations from live cell imaging of bead-loaded macrophages (Figure 1). We also considered a model where the constituents of dead cells are shared by multiple macrophages. However the model generated by this assumption is not consistent with our experimental data. We have updated the manuscript to make this point clear (lines 22-27 in the supplementary material).

How do authors quantify number of dead cells vs bits of apoptotic material? I was not clear on this point.

Our algorithm does not distinguish between live and dead cells. However, to disguise between live/dead cells and extracellular debris, we set a threshold for the cell size. Although we do not observed large amounts of this debris, this threshold prevents it from being counted as a cell. We have revised the methods section to make this point clear (see line 99 of the supplementary material).

The following sentence is maybe the main point of the paper and is hidden in the Discussion:

"cannibalistic efferocytosis has been overlooked as a mechanism of substance accumulation". So the Intro should possibly focus on this theme more clearly.

We agree and we have revised the manuscript accordingly (see lines 23-24 of the main text).

The authors state (line 266) that

" macrophage division and emigration might be beneficial to the resolution of inflammation.."

This sentence needs to be qualified, as it could be misinterpreted.

We agree and we have revised the manuscript accordingly (see lines 275-283 of the main text)

Typos:

Suppl 1 sec 1.7: Fix:

Substituting equations (27) and (??)

Line 64:

observed that macrophages
observed than macrophages

Line 78

macrophages numbers

Thank you for pointing these typos out. We have now revised the manuscript.

Overall, we would like to thank both reviewers for their important contributions. We think that addressing the reviewers comments has improved the our manuscript.